# Crystal Plasticity Finite Element Simulation of Grain Evolution Behavior in Aluminum Alloy Rolling

**DOI:** 10.3390/ma17153749

**Published:** 2024-07-29

**Authors:** Jun Li, Xiaoyan Wu, Haitao Jiang

**Affiliations:** 1National Engineering Research Center for Advanced Rolling and Intelligent Manufacturing, University of Science and Technology Beijing, Beijing 100083, China; cancan316@163.com (J.L.); wuxiaoyan@ustb.edu.cn (X.W.); 2NIO Technology (Anhui) Co., Ltd., Hefei 230001, China

**Keywords:** wrought aluminum alloy, crystal plasticity finite element, polycrystalline model, rolling deformation, texture

## Abstract

In this study, the crystal plasticity finite element method was established by coupling the crystal plasticity and finite element method (FEM). The effect of rolling deformation and slip system of polycrystalline Al-Mg-Si aluminum alloy was investigated. The results showed that there was a pronounced heterogeneity in the stress and strain distribution of the material during cold rolling. The maximum strain and shear strain occurred at surface of the material. The smaller the grain size, the lower the strain concentration at the grain boundary. Meanwhile, a smaller strain difference existed between the grain interior and near the boundary. The rotation of grains leads to significant differences in deformation and rotation depending on their initial orientations during the rolling process. The slip system of (11-1)<-110> had a large effect on the plastic deformation, (111)<10-1> is second, and the effect of (1-11)<011> slip system on the plastic deformation is the smallest. After deformation, the grain orientation concentration was increased with deformation. Therefore, both the strength and volume fraction of texture were increased with the increase in rolling deformation. The experimental results of EBSD indicated that the large rolling reduction resulted in severe grain twisting, so the texture strength was increased. The simulation results were in close agreement with the experimental results. This study provides a theoretical basis for the rolling process, microstructure, and performance control of aluminum alloys.

## 1. Introduction

Al-Mg-Si alloys are widely used to manufacture automobile body panels due to their excellent formability, corrosion resistance, and age-hardening properties [1,2]. The manufacturing process is the main bottleneck limiting the large-scale application of domestically produced automotive aluminum sheets in body panels, and the rolling process is the most important link in aluminum sheet manufacturing. During the rolling process, aluminum sheets undergo plastic deformation, resulting in a transition of most grains from isotropic to anisotropic. Numerous experiments have also shown that the mechanical behavior of materials, such as plastic deformation and fracture, is highly sensitive to crystal orientation [3].

The rolling of aluminum alloy sheet is a complex process, and various factors such as rolling temperature, rolling speed, and rolling deformation may affect the microstructure evolution, which in turn plays a crucial role in the mechanical properties of the metal sheet [4]. Meanwhile, the application of finite element simulation technology in the metal rolling process has become increasingly mature and widely used [5,6,7]. Compared with traditional experimental research methods, establishing a mathematical model based on deformation mechanism and microstructure evolution, using numerical simulation methods can more efficiently and accurately regulate the rolling process, and predict and improve the material microstructure during the rolling process.

The constitutive relationship of materials at the microscopic scale has been established to study the relationship between the microscopic and macroscopic properties of materials, which is a recent research hotspot in materials science and computational materials science [8]. Crystal plasticity-based finite element modeling (CP-FEM) combines the constitutive relationship of crystal plasticity with the finite element method. The CP-FEM constitutive model considered the plastic deformation of metals as generated by dislocation motion, slip, and lattice rotation. Compared with traditional constitutive relations, the CP-FEM model is closer to the physical essence of metal plastic deformation [9]. Therefore, it is of great significance to deeply understand the plastic deformation behavior and introduce the anisotropic deformation of grains into numerical simulation.

CP-FEM has been widely used in the study of the anisotropy and localization of material deformation. Lattice rotation has been considered to accurately capture the geometric softening effect, slip band evolution, and dislocation sliding in the deformation process. Numerous researchers have used the CP-FEM method to investigate many deformation processes of aluminum alloys, such as dynamic compression [10], plane stress tensile tests [11], asymmetric rolling [12], high-temperature [13], and low-temperature rolling [14], single crystal strength models [15], deformation-induced surface roughening [16], prediction of fatigue crack initiation [17]. In addition, this method has also been used to analyze the hot deformation behavior of Al-Li alloy [18].

In this study, the finite element numerical simulation method was employed to predict and analyze the microstructure of Al-Mg-Si alloy rolling process, study the rolling deformation resistance, stress–strain distribution, and grain orientation evolution law during the alloy rolling process, and analyze the influence of rolling deformation amount, slip system, grain size, and initial grain orientation on deformation behavior. The authors hope that the research results can provide a theoretical basis and reference for the optimization of the aluminum alloy rolling process and performance control.

## 2. Crystal Plasticity Model

### 2.1. Modeling Approach

The polycrystalline modeling method based on the Voronoi diagram has many advantages [19,20]. By controlling the position of Voronoi seed points, different shapes of polycrystalline sample models can be obtained. Then, by controlling random numbers, each Voronoi grain in the polycrystalline material is assigned a corresponding orientation and controlled to be random or some dominant orientation. This effectively describes the randomness of the size, shape, and spatial distribution of the microstructure of the material.

The finite element mesh may experience significant distortion, torsion, and mesh bending, among others, in the simulation of metal deformation processes. Therefore, this paper established a polycrystalline plastic finite element model based on the ABAQUS (V6.13)/INP file method, considering the characteristics of the rolling process. The schematic diagram of the process and steps for establishing a finite element model using ABAQUS/INP files is shown in Figure 1:(1)ABAQUS/CAE was used for simple modeling of computational models, and regular grids were used for grid partitioning. Then, the model nodes and unit information after grid division are exported and saved in a specific file format.(2)Based on the model size set in ABAQAUS/CAE, the number and orientation of grains are set in the MATLAB (R2021a) program, and a polycrystalline geometric model is established.(3)By calculating the coordinates of the cells and nodes generated in the first step, it was determined which grain they belonged to, and group the cells were grouped according to the grain.(4)The obtained cell information was written into the input file (INP file) of ABAQUS based on the grouping information of grains.(5)Compiling the information of each grain into the INP file and constructing a finite element model.

### 2.2. Polycrystalline Modeling

To generate the geometric data of aluminum Voronoi polycrystals in MATLAB (R2021a) software and write the data into the ABAQUS input file. The CAE pre-processing module of ABAQUS was used to obtain the geometric parameters of the crystal, and a two-dimensional model was constructed by combining the elastic coefficient, grain orientation, hardening parameters, and other parameters of the crystal. A three-dimensional polycrystalline geometric model was constructed using the INP file grain modeling method, and the Bassani–Wu hardening formula was used in the model [21]:(1)$$h_{\alpha \alpha }=F(\gamma^{\alpha })G(\gamma^{\beta })$$
(2)$$h_{\alpha \beta }=qh_{\alpha \alpha }$$
(3)$$F(\gamma^{\alpha })=(h_{0}-h_{s})sech^{2}\left[\frac{(h_{0}-h_{s})\gamma^{\alpha }}{\tau_{s}-\tau_{0}}\right]+h_{s}$$
(4)$$G(\gamma^{\beta })=1+\sum_{\beta =1,\beta \neq \alpha }^{n}\;f_{\alpha \beta }tanh(\gamma^{\beta }\gamma_{0})$$
where$F\left(\gamma^{\alpha }\right)$ represents the hardening modulus of the slip system;G represents potential hardening;$h_{0}$ represents the hardening coefficient at which yield occurs;γ represents the cumulative shear strain of all slip systems;$\tau_{0}$ represents the critical value of the initial decomposing shear stress;$\tau_{s}$ represents the critical value of shear stress of saturation decomposition;$f_{\alpha \beta }$ represents the interaction coefficient between slip system α and slip system β. When α = β, $f_{\alpha \beta }$ is 1, and when α≠β, $f_{\alpha \beta }$ is 0.$h_{\alpha \beta }$ represents the slip hardening coefficient, which is used to represent the hardening effect of slip shear strain in slip system β on slip system α.

The different colors in the model represent grains with different orientations, and the model contains 11,648 nodes and 9450 elements, respectively, as shown in Figure 2.

### 2.3. Material Parameters

According to the UMAT (User-defined material mechanical behavior) subroutine framework of ABAQUS software, material parameter input should follow the INP file (Input File) format and establish material attribute cards in a certain order. The original material used in this article was an annealed Al-Mg-Si polycrystalline aluminum alloy. Its chemical compositions (wt.%) are as follows: 0.8Mg-1.0Si-0.7Cu-0.25Fe-0.25Mn-0.1Cr-0.094Nb-(bal.) Al. The material parameters used in the Al-Mg-Si alloy model are based on the data from the reference [22], as shown in Table 1.

Where the elastic constants are as follows:C11 represents elastic deformation along the crystal axis direction.C12 represents elastic deformation within the crystal plane.C44 represents shear deformation when the principal axis is along the crystal diagonal direction.

Hardening parameters are as follows:*n* represents the hardening index.$\tau_{0}$ represents the critical value of the initial decomposing shear stress.$\tau_{s}$ represents the critical value of shear stress of saturation decomposition.$h_{0}$ represents the hardening coefficient at which yield occurs.q Latent hardening description.

The definition of $\tau_{0},\tau_{s},h_{0},q$ were based on the above-mentioned Equations (1)–(4).

### 2.4. Finite Element Model

The polycrystalline model was imported into ABAQUS/Standard and set up as a three-dimensional finite element model, as shown in Figure 3. The rolling mill was a rigid body with a diameter of 30 mm. To ensure the convergence of the calculation, the roll roller is pressed down a certain distance to ensure effective contact before rolling. The friction coefficient between the rolling mill and the contact surface of the model is defined as 0.15.

The boundary conditions during rolling are as follows: the bottom surface of the specimen remains constrained, except for the constraints removed at the tail of the specimen, and except for the rotational constraint in the rolling direction (TD direction) of the rolls.

## 3. Results and Discussion

### 3.1. Current Strength of Rolling Aluminum Alloy

The current strength in all slip systems is a state variable associated with the solution, serving as an internal variable that describes the current strength of slip systems. It can effectively reflect the deformation resistance experienced during the plastic deformation process of crystals [18]. The greater the deformation resistance, the more difficult it was for the slip system to initiate. It is manifested macroscopically as work hardening. In this article, the UMAT user subroutine is used to calculate the current strength (SDV1) of polycrystalline aluminum under different deformations. The predicted results are shown in Figure 4.

It can be seen in Figure 4 that the current strength of the sample shows an uneven distribution during the rolling deformation process. When the deformation is small, the maximum strength is concentrated at the grain boundaries and surfaces due to the obstruction of plastic slip at the grain boundaries. As the amount of deformation increases, the density of dislocations increases, and a large deformation resistance is generated inside the grains. Dislocations are distributed near grain boundaries and inside the grains. The entanglement of the large number of dislocations hindered the dislocation slip and increased the deformation resistance [23].

### 3.2. Stress and Strain Distribution

The non-uniform deformation during plastic deformation can be reflected by analyzing the stress–strain relationship. Figure 5a–c shows the strain distribution cloud map of the polycrystalline model after rolling under different amounts of compression. From the figure, it can be seen that during the cold rolling of polycrystalline aluminum, the strain distribution at the grain scale is uneven, and the strain value at the surface of the rolling model is relatively large. This is because boundary conditions are set on the surface of the model during rolling simulation. In addition, as the deformation increases, the strain value at the surface of the model becomes larger and larger. There are also differences in the strain values of grains at the same position within the rolling model, and the strain values will increase as the rolling deformation increases, as shown in Figure 5d–f.

Figure 6 shows the stress distribution cloud map of the whole model and typical grain positions under different rolling pressures with initial grain orientation. From Figure 6a–c, it can be seen that during rolling deformation, stress concentration occurs at the grain boundaries of the model due to the different initial orientations of each grain. Figure 6d–f shows the stress distribution of the same grain at typical positions under different amounts of compression. As shown in the figure, the internal stress of the polycrystalline material exhibits an uneven distribution during rolling deformation. The stress values at the grain boundaries differ greatly, and the stress values show an upward trend with increasing compression. The stress after rolling is concentrated in the rolling direction at 45° and 135° zones, which is caused by the three-dimensional stress during the rolling process. In summary, during the cold rolling process, the stress concentration is more likely to occur at the grain boundaries of polycrystalline aluminum, and the degree of plastic deformation of each grain is also different, resulting in uneven deformation and grain rotation.

### 3.3. Shear Stress and Strain

Figure 7 shows the shear strain distribution cloud map of the model when the rolling reduction is 5%, 10%, and 15%. It can be seen that the degree of deformation of grains with different orientations and positions varies.

Figure 8 shows the distribution of grain shear strain at different positions under 5% compression. We can see that the shear stress of grains located at the edge positions increases, indicating that grains are more prone to slip and contribute more to plastic deformation. The interaction between grains in the center position of the model is stronger. Their deformation is limited and more difficult during the rolling process.

The stress and shear strain distribution of two different initial grain orientations (model A is a random orientation and model B is a rotating cubic orientation) at a 10% reduction is shown in Figure 9. The initial grain size is represented by the number of 50 grains assigned in the simulation model. The two polycrystalline models only have different grain orientations, but there are significant differences in the stress and shear strain of the models after rolling. Model A has a random orientation, and the non-uniformity of deformation is more pronounced compared to the model due to different grain orientations, while the shear strain of grains at different positions in model B is basically the same.

The distribution of rolling shear strain for different grain sizes is shown in Figure 10. The difference in grain size is distinguished by the number of grains assigned in the same model. The more the number of crystals, the smaller the grain size. The number of grains under the same model are 20 (Figure 10a), 50 (Figure 10b) and 100 (Figure 10c), respectively. The deformation of grain-size materials has an effect, and the smaller the grain size, the lower the degree of strain concentration at the grain boundaries. At the same amount of deformation, the smaller the grain size, the more uniform the deformation distribution, and the smaller the difference in strain degree between the inside of the grain and the vicinity of the grain boundary.

### 3.4. Shear Stress and Strain in Different Slip Systems

The slip of dislocations promotes the plastic deformation of crystals, and the initiation of slip is the result of its shear stress effect [23]. In this study, the plasticity of polycrystalline aluminum cold-rolled crystals is simulated and the shear strain in different slip systems is output using the UMAT subroutine calculation in ABAQUS software. The SDVn represents the output of current strength under a certain state variable. The current strength SDV13-SDV24 in the simulation results corresponds one-to-one with a specific slip system.

When the rolling reduction is 10%, the distribution cloud map of the shear strain of various slip systems in polycrystalline aluminum is shown in Table 2. There are certain differences in the distribution of slip shear strain for each slip system, and the overall strain distribution is uneven. Aluminum is a typical face-centered cubic structure with 12 potential slip systems. This article focuses on the selection of S2, S7, and S12 slip systems to study their effects on the cold-rolling plastic deformation of polycrystalline aluminum.

To analyze the contribution of slip systems to plastic deformation in polycrystalline models, a statistical analysis of the cumulative shear strain of different slip systems was performed. The results are shown in Figure 11, where Figure 11a–c correspond to slip systems S2, S7, and S12, respectively. By comparing its strain distribution and maximum strain value, the result shows that slip system S12 has the highest cumulative shear strain, indicating a greater contribution to plastic deformation. The cumulative shear strain of the slip system corresponding to slip system S7 is the smallest, indicating a smaller contribution to plastic deformation. Therefore, in the cold rolling process of polycrystalline aluminum, S12 (11-1)<-110> slip has a large contribution to plastic deformation, followed by S2 (111)<10-1>, while the S7 (1-11)<011> slip system is not conducive to plastic deformation, and its contribution to plastic deformation is relatively small.

Figure 12 shows the cumulative shear stress distribution cloud map for different slip systems, where Figure 12a–c correspond to slip systems S3, S7, and S12, respectively. Similar to the distribution characteristics of shear strain, there is a significant uneven distribution of shear stress at the grain scale among the three slip systems during the cold rolling process. By comparing the shear stress values of the three slip systems, it can be concluded that the cumulative shear stress of the S12 slip system is greater than that of the S2 and S7 slip systems, and the grains are subjected to greater stress during rotation.

### 3.5. Influence of Rolling Deformation on Grain Orientation

Figure 13 shows the Scattering pole diagram of grain orientation changes after rolling with different reduction amounts, which was drawn using the common software MATLAB toolbox MTEX5.7. From the graph, we can see that new orientation poles appeared after rolling deformation. The orientation concentration increased and the grain orientation unfolded into more poles. The grains undergo rotation during the deformation process. As the rolling reduction increases, the density of orientation poles increases and the degree of grain rotation increases.

Three different initial orientations of grains were assigned to the finite element model, namely models A, B, and C (with random orientation, rotating cubic orientation, and cubic orientation, respectively). The {111} pole plots after rolling with different initial orientations are shown in Figure 14a–c. During the rolling deformation process, the grains rotate, and the initial orientation of the grains is different, resulting in significant differences in their deformation and rotation. Model B grains mainly rotate along the Z-axis direction, while model C grains mainly rotate along the X-axis direction with smaller orientation dispersion. This is mainly due to the uneven plastic deformation of different grains, resulting in uneven deformation and rotation of grains.

When assigning grain orientation, the algorithm produces a random orientation that follows a uniform distribution. The orientation distribution pole diagram of the orientation distribution of polycrystalline aggregates containing different numbers of grains is shown in Figure 15. The grains are distributed in all directions with no preferred orientation, and as the number of grains increases, the proportion of orientations in different intervals becomes more uniform.

Two different initial orientations were assigned to the finite element models B and C (cubic orientation and rotational cubic orientation, respectively). The {001} {101} and {111} pole plots are shown in Figure 16a,b. During the rolling deformation process, the grains will rotate, and the initial orientation of the grains is different, resulting in significant differences in their deformation and rotation. The grains of model B rotate mainly along the Z-axis direction, while the grains of model C rotate mainly along the X-axis direction with smaller orientation dispersion. This is mainly due to the uneven plastic deformation of different grains, resulting in uneven deformation and rotation of grains.

### 3.6. Experimental Study on Grain Characteristics after Cold Rolling

The process of experimental research is as follows:The rolling process involved hot rolling followed by cold rolling. After homogenizing, the block was heated in a box resistance furnace to 550 °C for 2 h, then underwent multi-pass rolling. The starting rolling temperature was 550 °C, and the final rolling temperature should not be lower than 300 °C. The block was rolled to produce sheets approximately 120 mm wide and with a thickness reduced from 60 mm to 5 mm. After rolling, the sheets were air-cooled to room temperature.The cold rolling process involved taking the 5 mm thick hot-rolled sheet and cold rolling it in 5 passes at room temperature to produce thin cold-rolled sheets approximately 130 mm wide and 1mm thick.The cold-rolled sheets were processed into rectangular plates measuring 220 mm × 70 mm. They underwent solution treatment at 550 °C for 25 min followed by quenching in water, and then underwent aging treatment in a resistance furnace.

The grain orientation distribution of Al-Mg-Si aluminum alloy with two different cold rolling reductions (40% and 80%) is shown in Figure 17. Both types of textures under compression show typical rolling textures, mainly including copper texture, brass texture, and S texture. However, the amount of deformation has a significant effect on the texture strength. When the deformation is small (40%), the maximum strength is only 11.4, and when the deformation is large (80%), the strength value reaches 38, which is more than three times the original value. The finite element simulation results of crystal plasticity in this study are consistent with the experimental results. The larger the deformation, the stronger the grain torsion, and the higher the texture strength.

The Inverse Pole Figure (IPF) after cold rolling with 40% and 80% reduction is shown in Figure 18. As the IPF plot showed, the grains of the two types of reduction samples have a fiber distribution that elongates along the rolling direction, and the grain orientation trend is approximately the same. However, with a larger reduction, the grain orientation distribution of the plate is stronger (C, D plots). The simulation results are consistent with the above-mentioned evolution law of grain orientation caused by rolling deformation.

## 4. Conclusions

In this study, a finite element simulation of the crystal plasticity of polycrystalline aluminum rolling was performed. The main conclusions are as follows.

(1)At the grain scale, there was a significantly uneven distribution of strain in polycrystalline aluminum cold rolling, and the stress value showed an upward trend with the increase in compression. The strain and shear strain at the surface of the model were larger than those inside, which made a larger contribution to plastic deformation. The interaction between grains in the central position was stronger, and the deformation was limited during the rolling process. The smaller the grain size, the lower the strain concentration at the grain boundary.(2)The grains were rotated and the initial orientation of the grains was different, resulting in significant differences in deformation and rotation during the rolling deformation process. In the cold rolling process of polycrystalline aluminum, S12 (11-1) <-110> slip had a larger contribution to the plastic deformation.(3)After rolling deformation, new orientation poles appeared and the concentration of orientation increased. The orientation of the grains expanded into more poles, and the grains rotated. The strength of texture and the volume fraction of texture increased with the increase in deformation.

## Figures and Tables

**Figure 1 materials-17-03749-f001:**
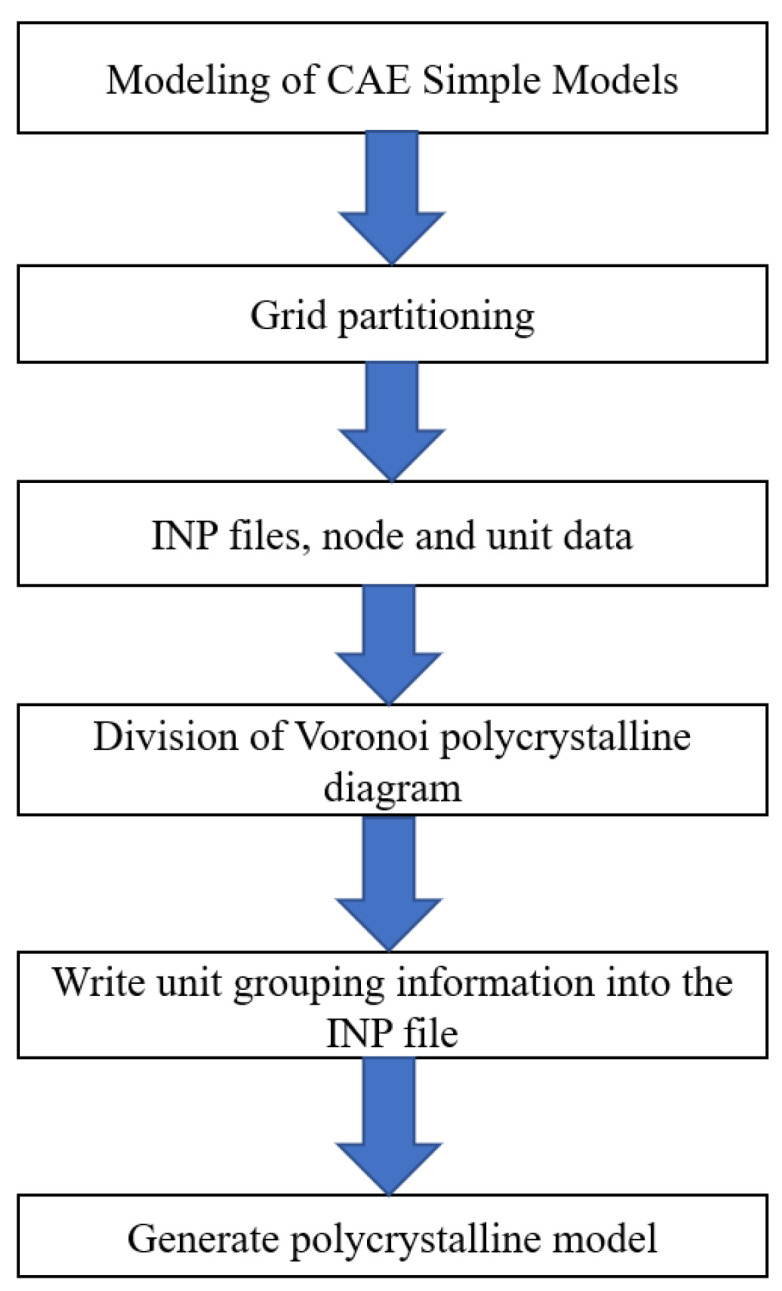
Schematic diagram of the process of building a polycrystalline model.

**Figure 2 materials-17-03749-f002:**
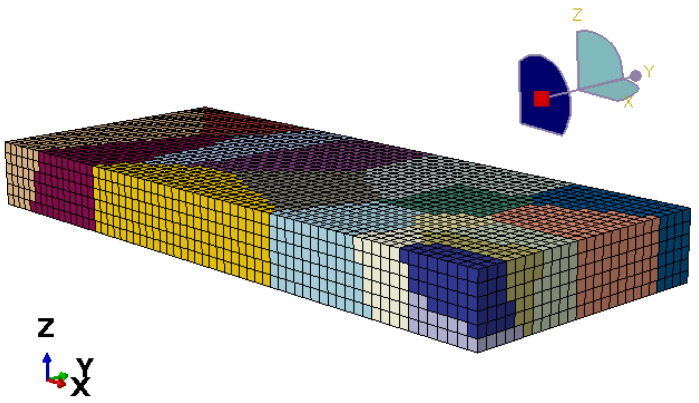
Polycrystalline model of rolled specimens.

**Figure 3 materials-17-03749-f003:**
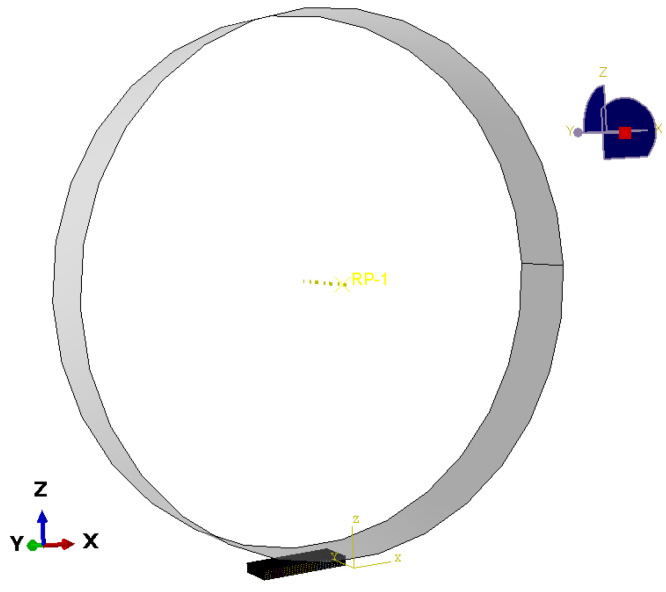
Finite element model of polycrystalline rolling.

**Figure 4 materials-17-03749-f004:**
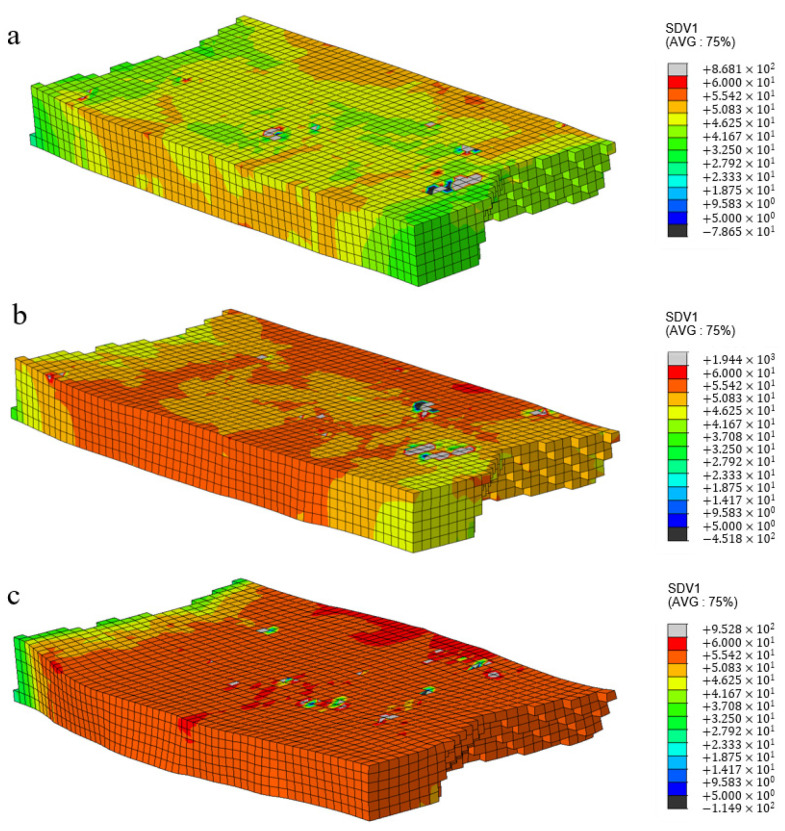
Current strength distribution of polycrystalline aluminum under rolling reduction at 5% (**a**), 10% (**b**), and 20% (**c**).

**Figure 5 materials-17-03749-f005:**
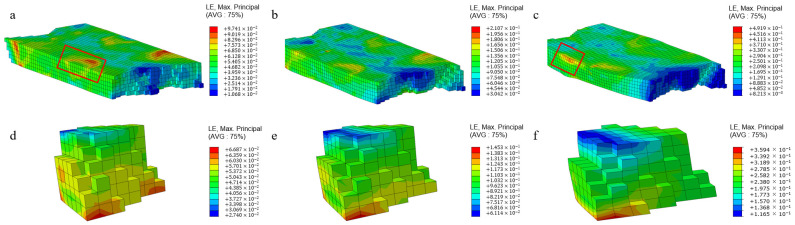
Strain distribution of polycrystalline aluminum during cold rolling under different rolling reduction: model strain distribution under rolling reduction at 5% (**a**), 10% (**b**) and 20% (**c**); strain distribution at the same position under rolling reduction at 5% (**d**), 10% (**e**) and 20% (**f**).

**Figure 6 materials-17-03749-f006:**
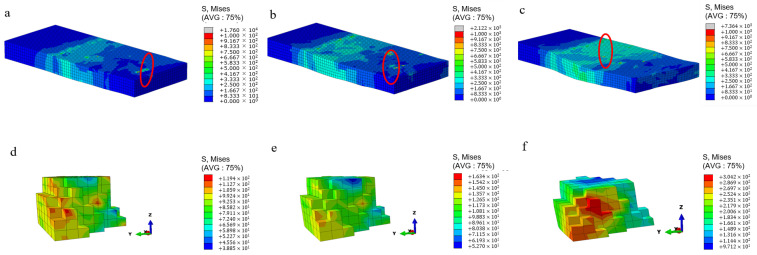
Stress distribution of polycrystalline aluminum during cold rolling under different rolling reduction: Model stress distribution under rolling reduction at 5% (**a**), 10% (**b**) and 20% (**c**); Typical grain stress distribution under rolling reduction at 5% (**d**), 10% (**e**) and 20% (**f**).

**Figure 7 materials-17-03749-f007:**
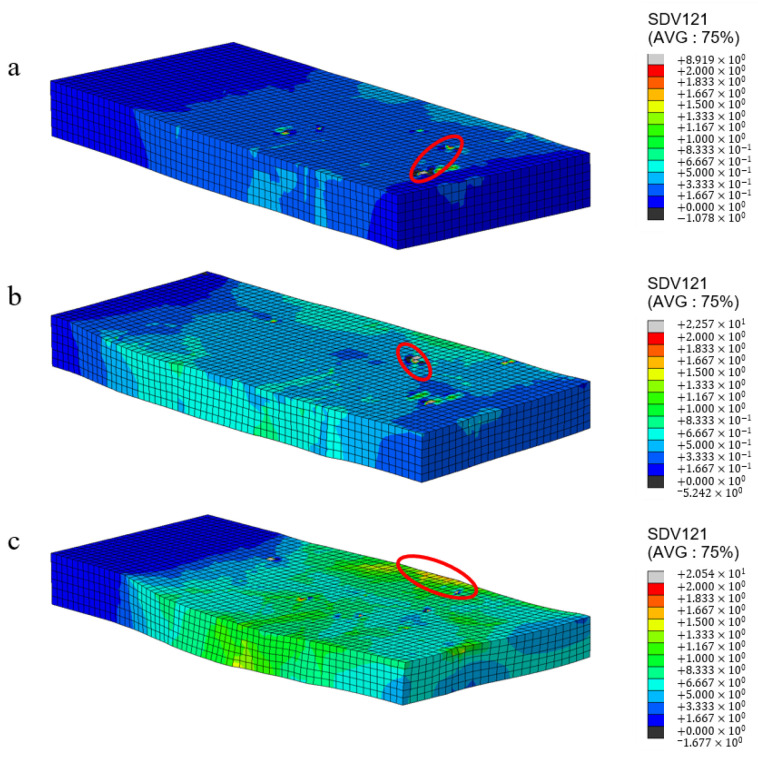
Distribution of shear strain under different rolling reductions at 5% (**a**), 10% (**b**) and 20% (**c**).

**Figure 8 materials-17-03749-f008:**
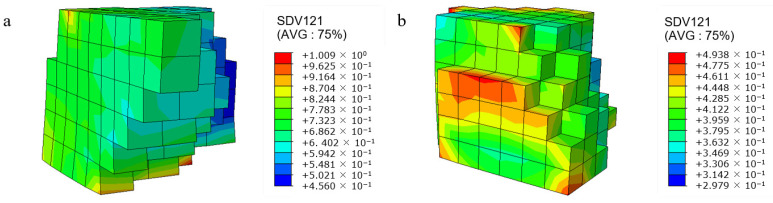
Shear strain distribution under 5% rolling reduction in the model: (**a**) at the edges; (**b**) inside.

**Figure 9 materials-17-03749-f009:**
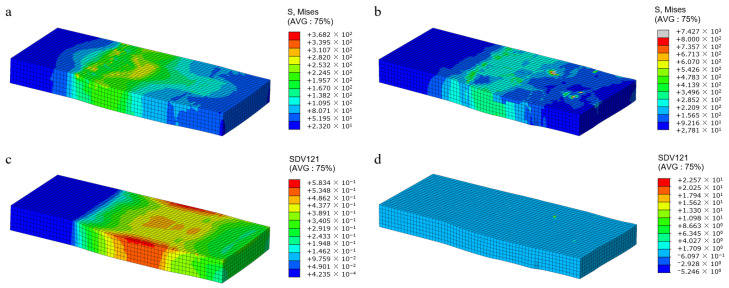
Stress and strain distribution during the rolling process with different initial orientations: (**a**,**c**) model A with random orientation; (**b**,**d**) model B with rotated cube orientation; (**a**,**b**) Mises stress; (**c**,**d**) shear strain.

**Figure 10 materials-17-03749-f010:**
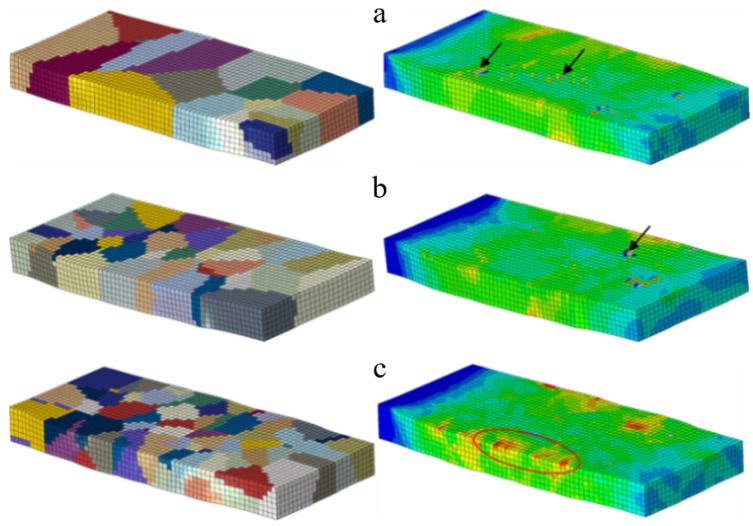
Shear strain distribution of different grain sizes: (**a**) 20-grain set; (**b**) 50-grain set; (**c**) 100-grain set.

**Figure 11 materials-17-03749-f011:**
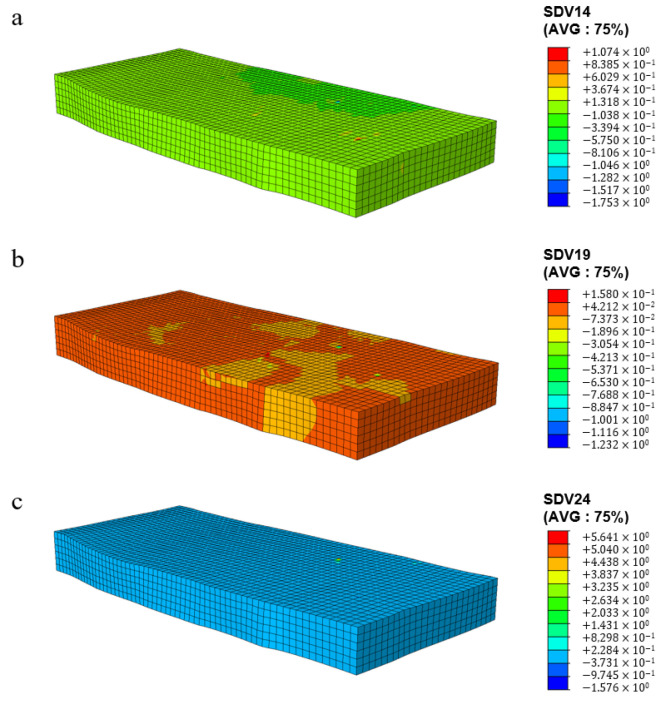
Shear strain distribution of different slip systems under 10% rolling reduction: (**a**) S2; (**b**) S7; (**c**) S12.

**Figure 12 materials-17-03749-f012:**
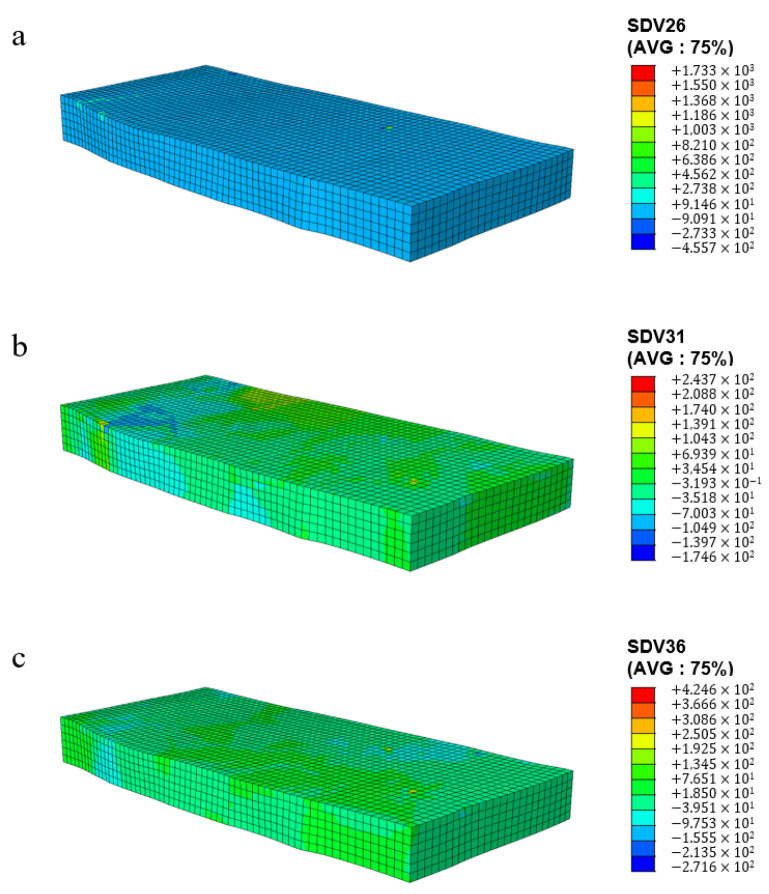
Shear stress distribution of different slip systems at 10% rolling reduction: (**a**) S2; (**b**) S7; (**c**) S12.

**Figure 13 materials-17-03749-f013:**
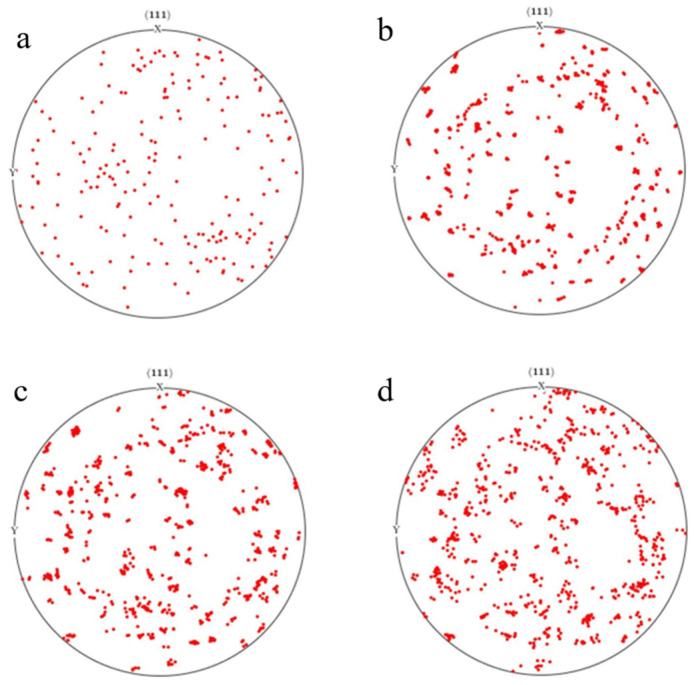
Scattering pole plots of rolling orientation change at different reduction levels: (**a**) initial orientation; (**b**) 5%; (**c**) 10%; (**d**) 20%.

**Figure 14 materials-17-03749-f014:**
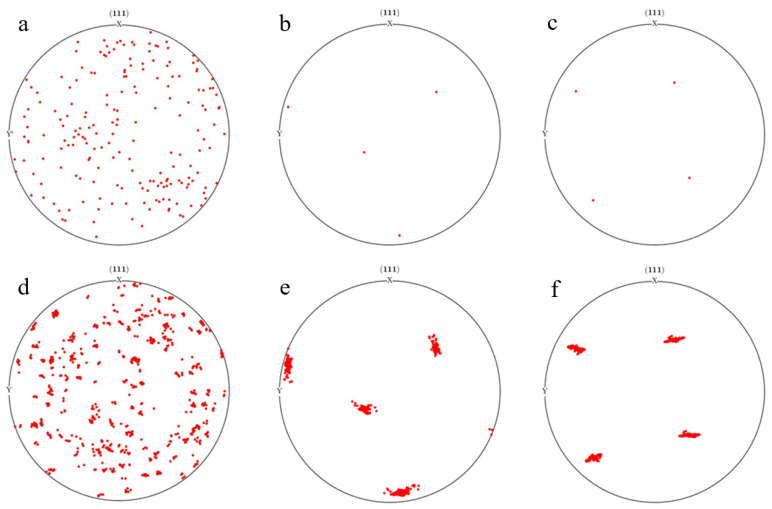
The {111} pole plots under different initial orientations: (**a**,**d**) model A with random orientation; (**b**,**e**) model B with rotating cubic orientation; (**c**,**f**) model C with cubic orientation.

**Figure 15 materials-17-03749-f015:**
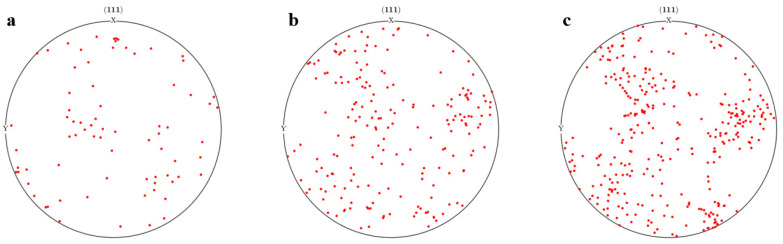
Polarization distribution of orientation for different numbers of grains: (**a**) 20; (**b**) 50; (**c**) 100.

**Figure 16 materials-17-03749-f016:**
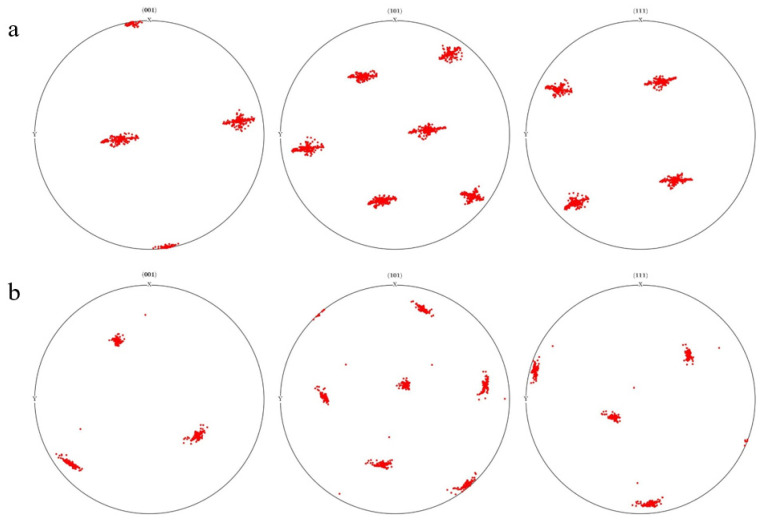
Pole plots of different initial orientation models: (**a**) model B with cubic orientation; (**b**) model C with rotational cubic orientation.

**Figure 17 materials-17-03749-f017:**
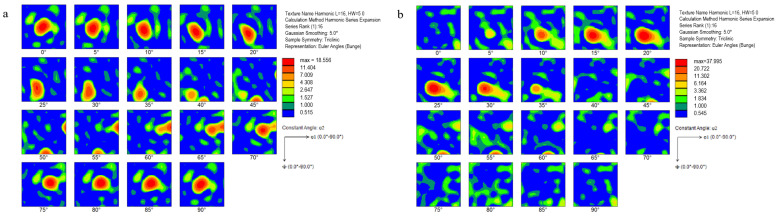
Orientation distribution of two cold rolling reduction values ODF: (**a**) small deformation; (**b**) large deformation.

**Figure 18 materials-17-03749-f018:**
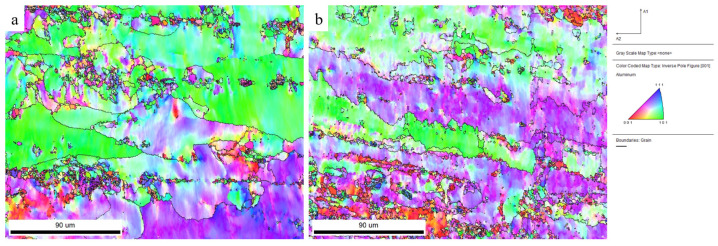
IPF plots of the model with two cold rolling reductions: (**a**) small deformation; (**b**) large deformation.

**Table 1 materials-17-03749-t001:** Crystal plasticity parameters of Al-Mg-Si Alloy for the Bassani–Wu model [21].

C11/MPa	C12/MPa	C44/MPa	n	q	$\dot{\gamma }(s^{-1})$	$h_{0}$/MPa	$\tau_{0}$/MPa	$\tau_{s}$/MPa
108,200	61,300	28,500	20	1.1	1	96	23.5	54

**Table 2 materials-17-03749-t002:** The UMAT output results of each slip system and the corresponding relationship with shear strain.

Serial Number of Sliding System	UMAT Component	Sliding System	Shear Strain
S1	SDV13	(111)<0-11>	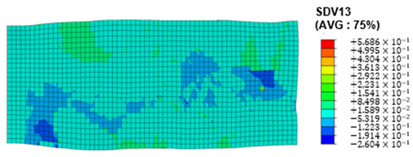
S2	SDV14	(111)<10-1>	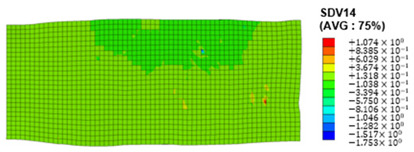
S3	SDV15	(111)<-110>	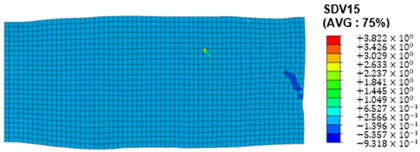
S4	SDV16	(-111)<101>	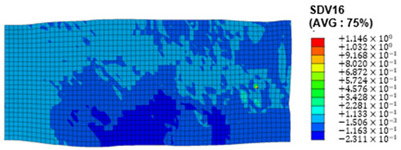
S5	SDV17	(-111)<110>	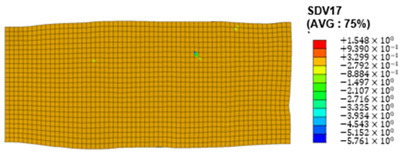
S6	SDV18	(-111)<0-11>	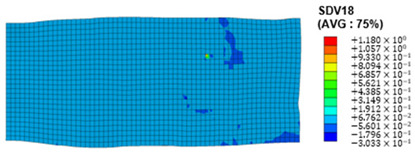
S7	SDV19	(1-11)<011>	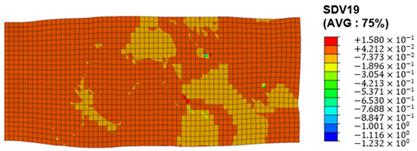
S8	SDV20	(1-11)<110>	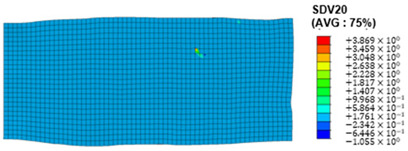
S9	SDV21	(1-11)<10-1>	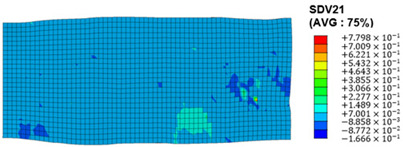
S10	SDV22	(11-1)<011>	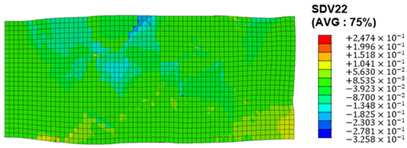
S11	SDV23	(11-1)<101>	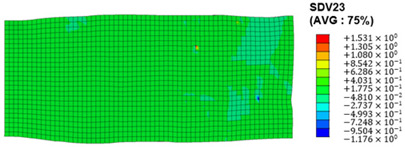
S12	SDV24	(11-1)<-110>	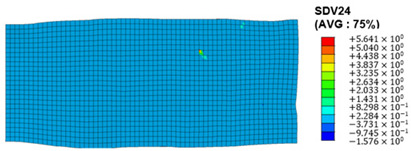

## Data Availability

The original contributions presented in the study are included in the article, further inquiries can be directed to the corresponding author.

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
