# Peer review of "Crystal Plasticity Finite Element Simulation of Grain Evolution Behavior in Aluminum Alloy Rolling"

_materials, 2024, doi:10.3390/ma17153749_

Round 1

Reviewer 1 Report

Comments and Suggestions for Authors

Dear authors,  

I have read your fascinating paper entitled "Crystal plasticity finite element simulation of grain evolution behavior in aluminum alloy rolling", but it has been rated "Reject" for the following reasons.

[1] The paper does not seem as well written as a submitted one. It would be better to review the instructions for authors. https://www.mdpi.com/journal/materials/instructions

[2] The analysis results of the crystal plasticity finite element method discuss the strain concentration at grain boundaries and within grains, but it is believed that the effect of grain boundaries cannot be determined from the analysis results. In addition, the analysis results also discuss dislocations within crystals, but while the existence of dislocations can be understood from the perspective of strain concentration, it does not feel that the distribution of dislocations at grain boundaries and within grains is directly shown.

The following are other points that need to be addressed, and we hope you will use them as a reference when making revisions.

[1] Reference [1], [4], [5], [19]: This book could not be verified. Please recheck the bibliographic information.

[2] P. 3 L. 106 "UMAT": Since this abbreviation is used for the first time in the main text, the full spelling should be given.

[3] p. 3 Table 1 "BW Model": We believe that this abbreviation should be defined in the main text (e.g., Bassani-Wu model: BW model).

[4] p. 3 "Bassani-Wu hardening formula": I think this formula should be shown in the main text.

[5] P. 3 "Table 1": I think the explanation of the variables in Table 1 should be given in the main text.

[6] P. 4 "Current Strength": This term seems difficult to understand. I think it would be better to change it to another expression to clarify what is meant by "current strength".

[7] p. 4 Figure 4: It is not clear from Figure 4 whether plastic deformation at grain boundaries is prevented. I think it would be more convincing if the actual grain boundaries were shown in the figure. Also, I don't think it is possible to determine the dislocation distribution from Figure 4.

[8] p. 5 Figures 5 and 6: It seems impossible to determine from Figures 5 and 6 that stress and strain are concentrated at grain boundaries. It would be more convincing if the actual grain boundaries were shown in the figures. Also, I think it would be better to place Figure 6 after the explanation of Figure 6 in the main text.

[9] p. 6 "Grain size is 50": What is the unit of crystal grain size given here?

[10] P. 7 "SDV13-SDV24": Please explain in the main text what SDV13 etc. means here.

[11] "References": Please review the instructions for authors and provide a list of references.

[12] "Other": Other points raised have been noted directly in the manuscript. We hope you will find them useful in your revisions.

Best regard,

Comments on the Quality of English Language

Author Response

Comments 1: [The paper does not seem as well written as a submitted one. It would be better to review the instructions for authors. https://www.mdpi.com/journal/materials/instructions]

Response 1: Thank you very much for your question. The author has made modifications and improvements according to the template format

Comments 2: [The analysis results of the crystal plasticity finite element method discuss the strain concentration at grain boundaries and within grains, but it is believed that the effect of grain boundaries cannot be determined from the analysis results. In addition, the analysis results also discuss dislocations within crystals, but while the existence of dislocations can be understood from the perspective of strain concentration, it does not feel that the distribution of dislocations at grain boundaries and within grains is directly shown.

Response 2: Thank you very much for your question.

When generating geometric data for aluminum plates using Voronoi diagrams, the grain boundaries are determined, which is related to the definition of Voronoi diagrams.

The output results from ABAQUS can be visualized using post-processing software as contour plots, various cloud charts, etc. These results may include stress and strain, displacement and deformation, temperature and frequency, among others. But the dislocation distribution may require further processing of the results using other software. The evolution of dislocations during the rolling process is not the focus of this article, so a detailed study of dislocations has not been conducted.

Comments 3: [Reference [1], [4], [5], [19]: This book could not be verified. Please recheck the bibliographic information. ]

Response3: Thank you very much for your question. The author has searched for new references and replaced them.

Comments 4:  P. 3 L. 106 "UMAT": Since this abbreviation is used for the first time in the main text, the full spelling should be given. ]

Response 4: Thank you very much for your question. The User defined Material Mechanical Behavior (UMAT) subroutine in the UMAT user subroutine, User-defined mechanical material behavior, and also annotated in the paper.

Comments 5: [p. 3 Table 1 "BW Model": We believe that this abbreviation should be defined in the main text (e.g., Bassani-Wu model: BW model). ]

Response 5:

Thank you very much for your question. The author has changed the title to 'Bassani Wu model'

Comments 6: ["Bassani-Wu hardening formula": I think this formula should be shown in the main text. ]

Response 6: Thank you very much for your question.

Bassani and Wu's hardening model:

Reference:

Bassani, J.L. and T.-Y. Wu, Latent Hardening in Single Crystals II. Analytical Characterization and Predictions. Proceedings of the Royal Society of London. Series A: Mathematical and Physical Sciences, 1991. 435(1893): p. 21–41.

Comments 7: [ "Table 1": I think the explanation of the variables in Table 1 should be given in the main text. ]

Comments 8: [ P. 4 "Current Strength": This term seems difficult to understand. I think it would be better to change it to another expression to clarify what is meant by "current strength".]

Response 8: Thank you very much for your question.

The current strength in all slip systems is a state variable associated with the solution, serving as an internal variable that describes the current strength of slip systems. (The current strength means the present strength, not the electrical current strength)

Comments 9: [ p. 4 Figure 4: It is not clear from Figure 4 whether plastic deformation at grain boundaries is prevented. I think it would be more convincing if the actual grain boundaries were shown in the figure. Also, I don't think it is possible to determine the dislocation distribution from Figure 4. ]

Response 9: Thank you very much for your question.

The generation of dislocations is one of the mechanisms of deformation strengthening in metal materials: as plastic deformation progresses, the density of dislocations continuously increases. Therefore, the mutual delivery of dislocations during movement intensifies, increasing the resistance to dislocation movement and causing an increase in deformation resistance, thereby improving strength. Dislocations are not the simulation results presented in this article, but rather an analysis of the metal deformation strengthening mechanism based on the displayed results. Relevant references were also cited.

Comments 10: [p. 5 Figures 5 and 6: It seems impossible to determine from Figures 5 and 6 that stress and strain are concentrated at grain boundaries. It would be more convincing if the actual grain boundaries were shown in the figures. Also, I think it would be better to place Figure 6 after the explanation of Figure 6 in the main text. ]

Response 10: Thank you very much for your question.

The concentration of strain and strain can be preliminarily judged from Figs.  6d-f, but this article done not have a relatively accurate quantitative measurement of stress and strain.

The author adjusted the order of Figure 6 and its explanation in the main text.

Comments 11: [ p. 6 "Grain size is 50": What is the unit of crystal grain size given here? ]

Response 11: Thank you very much for your question.

Here, a grain size parameter is assigned to the model to compare the effects of different initial grain orientations under a certain grain size. The initial grain size is represented by the number of 50 grains assigned in the simulation model. It is not the actual size of the grains.

Comments 12: ["SDV13-SDV24": Please explain in the main text what SDV13 etc. means here. ]

Response 12: Thank you very much for your question.

SDVn represents the output of current strength under a certain state variable. The current strength SDV13-SDV24 in the simulation results corresponds one-to-one with a specific slip system.

SDV          for all solution dependent state variables 

SDVn           for the solution dependent state variable n

Comments 13: [ "References": Please review the instructions for authors and provide a list of references. ]

Response 13: Thank you very much for your question. The author has updated the citation of the references as required.

Comments 14: ["Other": Other points raised have been noted directly in the manuscript. We hope you will find them useful in your revisions. ]

Response 14: Thank you very much for your question.

Thank you very sincerely for your patient editing and hard work. I have carefully read and revised it.

Reviewer 2 Report

Comments and Suggestions for Authors

What is the novely of the paper?

Extend the introduction background.

Line 101 – what means units in the model?

Check size of columns in tab. 1.

Line 118 – Friction coefficient with unit?

How the current strength (fig. 4) is calculated?

Figure 5 -  the LE is logarithmic strain – this strain consist from elastic and plastic part or?

Add more information about FE mesh, model, speed of rotation, boundary condition.

Author Response

Dear Reviewers:
We have studied the comments carefully and tried our best to revise the manuscript in detail. The grammar of revised manuscript also has been proofed by professional. Revised portion is marked in red in the marked up manuscript. The point to point responses to your comments and the explanations regarding our revisions are listed in the text below. With these revisions, we hope our manuscript could meet standards of materials journal and will be acceptable for publication. You can see in attachment.

We appreciate for your warm work earnestly. Please feel free to contact us with any questions and we are looking forward to hearing from you soon.

Best regards

Jun Li

Reviewer 3 Report

Comments and Suggestions for Authors

Research using CP-FEM modeling is crucial for deeper understanding and prediction of the behavior of metal materials. They have a direct impact on the development of production technologies, process optimization and innovation in the field of new materials. However, there are several important research problems that require further analysis, such as: investigation of the mechanisms leading to irregular distribution of deformation and ways of minimizing them by optimizing the parameters of the rolling process; expansion of the research into different temperature ranges to understand the effect of heat on plastic distortion processes, especially for materials used in variable temperature conditions. The EBSD results were well interpreted, showing that a greater reduction in rolling leads to a stronger rotation of the grains mainly due to increased plastic deformation, increased number of dislocations, voltage gradients and changes in the crystallographic texture. All these factors affect the microstructure of the alloy  and cause the grains to twist.

Notes

1. In the sentence marked 66 remove the slide when quoting literaturÄ™

2. In table 1, adjust the width of the last column so that the unit of tension is readable

3. Change the paragraph so that the sentence starts with the main letter:

Fig. 7 shows the shear strain distribution cloud map of the model when the rolling reduction is 5%, 10%, and 15%. It can be concluded that the degree of deformation of grains with different orientations and positions during the rolling process varies. the increase in deformation, the non-uniformity of deforming increases, and the shear stress on the edges changes greatly.

4. Correction of signature under Fig. 8. The passage concerning Figure 9 should begin with a separate paragraph in the text.

5.  The sentence marked  193: the initial grain size without a unit is indicated. In addition, the signature under Figure 10 contains information about the grain size of the harvest, not the grains size - please unify.

6. In the sentence marked 335 should be.. with the increase in deformation. 

Author Response

Dear Reviewers:
We have studied the comments carefully and tried our best to revise the manuscript in detail. The grammar of revised manuscript also has been proofed by professional. Revised portion is marked in red in the marked up manuscript. The point to point responses to your comments and the explanations regarding our revisions are listed in the text below. With these revisions, we hope our manuscript could meet standards of materials journal and will be acceptable for publication. You can see in attachment.

We appreciate for your warm work earnestly. Please feel free to contact us with any questions and we are looking forward to hearing from you soon.

Best regards

Jun Li

Comments 1: [In the sentence marked 66 remove the slide when quoting literature]

Response 1: Thank you very much for your question. The authors have reviewed and revised the paper.

Comments 2: [ In table 1, adjust the width of the last column so that the unit of tension is readable]

Response 2: Thank you very much for your question. The authors have made corresponding adjustments in the paper

Comments 3: [Change the paragraph so that the sentence starts with the main letter:(Fig. 7 shows the shear strain distribution cloud map of the model when the rolling reduction is 5%, 10%, and 15%. It can be concluded that the degree of deformation of grains with different orientations and positions during the rolling process varies. the increase in deformation, the non-uniformity of deforming increases, and the shear stress on the edges changes greatly.)]

Response 3:Thank you very much for your question. The authors have modified the relevant description.

Comments 4: [Correction of signature under Fig. 8. The passage concerning Figure 9 should begin with a separate paragraph in the text. ]

Response 4: Thank you very much for your question. The authors have made revisions in the paper

Comments 5: [The sentence marked 193: the initial grain size without a unit is indicated. In addition, the signature under Figure 10 contains information about the grain size of the harvest, not thgrains size - please unify. ]

Response 5: Thank you very much for your question. The authors have made revisions in the paper

Comments 6: [In the sentence marked 335 should be.. with the increase in deformation ]

Response 6: We have made revisions to the paper.

Reviewer 4 Report

Comments and Suggestions for Authors

The paper Crystal plasticity finite element simulation of grain evolution  behavior in aluminum alloy rolling” is a theoretical research paper, fit to SI topics.

The Abstract has to be improved, so that to evidence the scope and objectives of research (in its first part) and to mention further research development (by the end of it).

There are major English errors, as for example: “The effect of rolling deformation and slip system  of polycrystalline Al-Mg-Si aluminum alloy on deformation resistance, stress-strain distribution, and grain orientation evolution during rolling process.” that lacks the predicate. (L9 - L11).

In the last paragraph of Introduction, I suggest to mention why / the reason and importance of the study (L59 - 61). Also, if any similar studies and associated results have been published.

Please pay attention to formulation: “this paper established a polycrystalline plastic finite element” (L74) as, the authors usually “establish” .

Rephrase and use explicit sentences for a research articles - see for axample the formation "Generate geometric data of aluminum Voronoi polycrystals in MATLAB software  and wrote the data into the ABAQUS input file. Utilize the pre-processing module CAE  of ABAQUS to obtain the geometric parameters of the crystal (L94 - L96)”.

For the moment it sounds like a sequence of procedure information !!!!

The units for friction coefficient are non-dimensional. Explain why there is mentioned  0.15 N/m2.

At L255 and beyond, define orientation changes - whose orientation ?

The paper have to be written according to Materials journal template. This is why, please reconfigure the paper as mentioned by the template - especially, chapter 2.

Please, IF POSSIBLE, describe the (real) experimental system mentioned in subchapter 3.6. (L296, L304).

In last chapter, Conclusion, please mention further research development.

MAJOR English check is, required

Comments on the Quality of English Language

Low quality of English.

Needs professional help.

Author Response

Comments 1: [ The Abstract has to be improved, so that to evidence the scope and objectives of research (in its first part) and to mention further research development (by the end of it). ]

Response 1: Thank you very much for your question.

The authors have improved the abstract, and clarified the objectives of optimization research and other expressions.

Comments 2: [ There are major English errors, as for example: “The effect of rolling deformation and slip system  of polycrystalline Al-Mg-Si aluminum alloy on deformation resistance, stress-strain distribution, and grain orientation evolution during rolling process.” that lacks the predicate. (L9 - L11). ]

Response 2: Thank you very much for your question. I have carefully read and revised it

Comments 3: [In the last paragraph of Introduction, I suggest to mention why / the reason and importance of the study (L59 - 61). Also, if any similar studies and associated results have been published. ]

Response 3 Thank you very much for your question.

The author has rewritten the introduction section, please review it in the revised version of the paper.

Comments 4: [Rephrase and use explicit sentences for a research articles - see for axample the formation "Generate geometric data of aluminum Voronoi polycrystals in MATLAB software  and wrote the data into the ABAQUS input file. Utilize the pre-processing module CAE  of ABAQUS to obtain the geometric parameters of the crystal (L94 - L96)”.For the moment it sounds like a sequence of procedure information !!!! ]

Response 4: Thank you very much for your question. The authors have made revisions in the paper

Comments 5: [ The units for friction coefficient are non-dimensional. Explain why there is mentioned  0.15 N/m2. ]

Response 5: Thank you very much for your question. The coefficient of friction does not have a unit.

Comments 6: [At L255 and beyond, define orientation changes - whose orientation ? ]

Response 6: Thank you very much for your question.

The orientation refers to the change in grain orientation after rolling. The author has added the attribute of grain in the corresponding position

Comments 7: [The paper have to be written according to Materials journal template. This is why, please reconfigure the paper as mentioned by the template - especially, chapter 2.  ]

Response 7: Thank you very much for your question. The author has revised the paper.

Comments 8: [Please, IF POSSIBLE, describe the (real) experimental system mentioned in subchapter 3.6. (L296, L304). ]

Response 8: Thank you very much for your question.

The process of experimental research is as follows:

The rolling process involves hot rolling followed by cold rolling. After homogenizing, the block is heated in a box resistance furnace to 550°C for 2 hours, then undergoes multi-pass rolling. The starting rolling temperature is 550°C, and the final rolling temperature should not be lower than 300°C. The block is rolled to produce sheets approximately 120mm wide and with a thickness reduced from 60mm to 5mm. After rolling, the sheets are air-cooled to room temperature.

The cold rolling process involves taking the 5mm thick hot-rolled sheet and cold rolling it in 5 passes at room temperature to produce thin cold-rolled sheets approximately 130mm wide and 1mm thick.

The cold-rolled sheets are processed into rectangular plates measuring 220mm × 70mm. They undergo solution treatment at 550°C for 25 minutes followed by quenching in water, and then undergo aging treatment in a resistance furnace.

Comments 9: [In last chapter, Conclusion, please mention further research development. ]

Response 1: Thank you very much for your question. The author has revised the paper.

Comments 10: [MAJOR English check is, required]

Response 10: Thank you very much for your question. The author has revised the paper.

Comments 11: [ Please pay attention to formulation: “this paper established a polycrystalline plastic finite element” (L74) as, the authors usually “establish”. ]

Response 11: Thank you very much for your question. The author has revised the paper.

Round 2

Reviewer 1 Report

Comments and Suggestions for Authors

Good!

Author Response

(The authors gave the same response as above.)

Reviewer 2 Report

Comments and Suggestions for Authors

I after with publication

Author Response

Comments 1: [What is the novely of the paper?]

Response 1: Thank you very much for your question.

The lightweigh of car body is a major trend in the future development of the automotive industry, and aluminum alloy is the preferred material for achieving lightweight for car body.

As we all known. analyzing the evolution of microstructure with process parameters solely through experimental methods is a complex task. Therefore, this article establishes a polycrystalline plastic finite element model based on the ABAQUS/INP file method through secondary development of ABAQUS. The proposed microstructure model can be used to predict and analyze the microstructure of Al-Mg-Si polycrystalline aluminum alloy during cold rolling process.

In addition, this article also conducted cold rolling experiments on Al-Mg-Si aluminum alloy, observed and analyzed the microstructure of the rolled alloy, and compared it with simulation results. The results indicate that the established mathematical model can accurately predict the microstructure of Al-Mg-Si aluminum alloy during rolling. To provide an effective reference method for optimizing the microstructure of aluminum alloy rolling process, and to obtain the influence of main process parameters on the microstructure of the rolling process, provide theoretical basis for optimizing the rolling process and performance control of aluminum alloy, and provide ideas for researchers with finite element simulation of other microstructures, and provide reference for the optimization of rolling processes in related industries in the future.

Comments 2: [Extend the introduction background. ]

Response 2: Thank you very much for your question.

The article already introduces automotive aluminum plates and crystal plastic finite elements, so the author added "Background of Al-Mg -Si alloy for automotive use" and reflected it in the background introduction.

Al-Mg-Si alloys are currently a hot topic in automotive materials research and applications, notable for their paint bake hardening characteristics. Their maximum attribute lies in achieving final strength only during the paint baking stage of the automotive structure, allowing them to exhibit good formability in the solution treatment state (T4 temper) and increased strength in the age-hardened state (T6 or T8 temper). Aluminum alloys from the Al-Mg-Si series commonly used for automotive body panels include AA6009, AA6010, AA6016, and AA6111.

Comments 3: [Line 101 – what means units in the model? ]

Response 3:

In the context of finite element simulation, an element refers to a cubic element formed by the mesh division shown in Fig. 2. In the paper, the “ellments” was used instead of “units”.

In finite element analysis, it is necessary to discretize a continuum (an infinite number of micro elements) using simple finite elements. Each element is defined with known properties, and then, based on certain assumptions (such as various assumptions in mechanics including continuity, linearity, compatibility, etc.), finite element modeling can be conducted.

Comments 4: [ Check size of columns in tab. 1. ]

Response 4:

Thank you very much for your question. The authors have made revisions in the paper

Comments 5: [Line 118 – Friction coefficient with unit? ]

Response 5: Thank you very much for your question. The coefficient of friction does not have a unit.

Comments 6: [How the current strength (fig. 4) is calculated? ]

Response 6: Thank you very much for your question.

The current strength in all slip systems is a state variable associated with the solution, serving as an internal variable that describes the current strength of slip systems. (The current strength means the present strength, not the electrical current strength). The author also provided supplementary explanations in the main body of the paper.

Comments 7: [Figure 5 -the LE is logarithmic strain – this strain consist from elastic and plastic part or? Add more information about FE mesh, model, speed of rotation, boundary condition. ]

Response 7:

Thank you very much for your question.

LE is indeed logarithmic strain, but in ABAQUS it usually refers to true strain,and this strain consist from elastic and plastic. The true stress is usually calculated by the following equation:

More information about the FE mesh were added in the paper:

The rolling mill rolls are rigid bodies with a diameter D = 30mm. During the simulation process, to ensure convergence of the calculation, the rolls are pressed down a certain distance to establish effective contact before rolling begins. The coefficient of friction between the rolls and the specimen contact surface is 0.15.

The boundary conditions during rolling are as follows: the bottom surface of the specimen remains constrained, except for the constraints removed at the tail of the specimen, and except for the rotational constraint in the rolling direction (TD direction) of the rolls.

Reviewer 4 Report

Comments and Suggestions for Authors

Please check at lines 84 - 86 and, similarly, 230 - 232

As there is a multiple subject (for example L 84-86: The stress and shear strain distribution (…) is are shown in Fig. 9.)

Please add further research development in Conclusion chapter

Comments on the Quality of English Language

Minor English editing required.

Author Response

(The authors gave the same response as above.)
